# Fabrication of Rechargeable Photoactive Silk Fibroin/Polyvinyl Alcohol Blend Nanofibrous Membranes for Killing Bacteria

**DOI:** 10.3390/polym14122499

**Published:** 2022-06-19

**Authors:** Shixiong Yi, Jiaxue Wu, Ying Zhou, Xiaomeng Wang, Yunfei Pu, Boli Ran

**Affiliations:** 1State Key Laboratory of Silkworm Genome Biology, College of Sericulture, Textile and Biomass Sciences, Southwest University, Chongqing 400715, China; 13193177884@163.com (J.W.); 17856104528@163.com (Y.Z.); wxm1361753159@163.com (X.W.); 2Key Laboratory of Textile Fiber and Products, Ministry of Education, Wuhan Textile University, Wuhan 430200, China; 3Department of Cardiovascular Medicine, Chongqing General Hospital, Chongqing 401147, China; puyunfei09@foxmail.com

**Keywords:** photoactive, silk fibroin, polyvinyl alcohol, nanofibrous membranes, killing bacteria

## Abstract

Antibacterial materials that prevent bacterial infections and mitigate bacterial virulence have attracted great scientific interest. In recent decades, bactericidal polymers have been presented as promising candidates to combat bacterial pathogens. However, the preparation of such materials has proven to be extremely challenging. Herein, photoactive silk fibroin/polyvinyl alcohol blended nanofibrous membranes grafted with 3,3’,4,4’-benzophenone tetracarboxylic dianhydride (G-SF/PVA BNM) were fabricated by an electrospinning technique. The premise of this work is that the G-SF/PVA BNM can store photoactive activity under light irradiation and release reactive oxygen species for killing bacteria under dark conditions. The results showed that the resultant G-SF/PVA BNM exhibited the integrated properties of an ultrathin fiber diameter (298 nm), good mechanical properties, robust photoactive activity and photo-store capacity, and great photoinduced antibacterial activity against *E. coli* and *S. aureus* (99.999% bacterial reduction with 120 min). The successful construction of blended nanofibrous membranes gives a new possibility to the design of highly efficient antibacterial materials for public health protection.

## 1. Introduction

Bacterial infections have been a major cause of disease throughout the history of human populations, and in recent years, have imposed a tremendous economic burden on patients [1,2,3,4]. Bacterial infection is one of the major causes of many deadly diseases and causes millions of deaths all over the world every year. To date, for the aim of antibacterial or bacteria-killing purposes, much effort has been focused on the development of new techniques [5,6,7,8]. The most efficient way is to construct a surface with photocatalytic substances, such as anatase titanium dioxide, magnesium oxide, and tin oxide, which could generate reactive oxygen species (ROS) for bioprotective applications [9,10,11,12]. However, the mixing of these nanoparticles with polymer matrixes often leads to poor affinity and low interactions between them [13,14,15,16]. Organic soluble photosensitive complexes having strong interactive groups with most polymers and textile materials are more appealing for textile applications towards the majority of organic fibers [17,18,19,20].

In previous reports, some polymeric nanofibrous membranes such as ethylene-vinyl alcohol copolymer, polyacrylonitrile, and poly (vinyl alcohol) were fabricated by an electrospinning technology for various applications. However, these polymers are not biodegradable, and the discarding of the functional polymers will lead to serious environmental concerns. Silk fibroin (SF) is one of the main abundant natural fibers, and it can be obtained simply and economically. The use of SF has been extended to various high-tech application areas including biomaterials and tissue engineering. In our previous study, we reported on photoactive SF nanofibrous membranes for the degradation of colored dyes [21]. In fact, some disadvantages of pure SF nanofibrous membranes were reported, such as being brittle, having tedious degradation rate control, and so on [22,23,24,25]. Thus, many silk fibroin-based blended nanofibrous membranes were constructed to enhance its performances in various applications [26,27,28,29]. Polyvinyl alcohol (PVA) deserves special attention for its remarkable characteristics such as hydrophilicity, biodegradability, elasticity, and biocompatibility [30,31,32,33].

Here, we fabricated recharged photoactive silk fibroin/polyvinyl alcohol blended nanofibrous membranes (G-SF/PVA BNM) grafted with 3,3’,4,4’-benzophenone tetracarboxylic dianhydride for killing *Escherichia coli* (*E. coli*) and *Staphylococcus aureus* (*S. aureus*), as shown in Figure 1. The results demonstrated the G-SF/PVA BNM exhibited an ultrathin fiber diameter, robust photoactive activity, excellent antibacterial capacity, good mechanical properties, and good reusability, and it could store photoactive activity under light irradiation and release ROS under dark conditions. The successful construction of blended nanofibrous membranes gives a new possibility to the design of highly efficient antibacterial materials for public health protection.

## 2. Experimental Details

### 2.1. Materials and Reagents

Silk cocoon was supplied from Southwest University (Chongqing, China). Polyvinyl alcohol (PVA), 3,3’,4,4’-benzophenone tetracarboxylic dianhydride (BTDA), polyphosphoric acid (PPA), tetrahydrofuran (THF), potassium iodide, ammonium molybdate tetrahydrate, dioxane, potassium hydrogen phthalate, formic acid, *p*-nitrosodimethylaniline (*p*-NDA), and all other chemicals were purchased from the Chongqing Siwei Laboratory Instrument Company (Chongqing, China).

### 2.2. Electrospinning of SF-PVA Nanofibers

Firstly, the silk cocoon was pretreated for removing sericin and further dialyzing using a method similar that of our previous work [34]. After that, the silk fibroin sponges and PVA were mixed to obtain the electrospinning solution. A voltage of 16 kV was applied to a droplet of the mixed solution at the tip (ID 0.495 mm) of a syringe needle. Finally, the electrospinning experiment was carried out at a controlled feed rate of 1.0 mL h^−^^1^. The silk fibroin/polyvinyl alcohol blended nanofibrous membranes (SF/PVA BNM) were collected on a target drum, which was placed at a distance of 12 cm from the syringe tip. 

### 2.3. Chemical Modification of Membranes

The PPA and BTDA were mixed and dissolved in dioxane. The G-SF/PVA BNM was obtained by immersing SF/PVA BNM in mixed dioxane. Finally, the resulting G-SF/PVA BNM was removed, washed, and dried. 

### 2.4. Measurement of Mechanical Properties

The mechanical properties of the membranes were measured using a tensile tester. Samples were resized at 10 mm length and then fixed between the machine’s grips at a crosshead speed of 5 mm/min.

### 2.5. Evaluation of the ROS

We employed OH• and H_2_O_2_ to evaluate the release capacity of the ROS. The concentration of OH• and H_2_O_2_ was detected according to the methods in our previous report [35].

### 2.6. Antibacterial Activity of G-SF/PVA BNM

The antibacterial activity was evaluated against *E. coli* and *S. aureus*. Firstly, the G-SF/PVA BNM with a size of 2 × 2 cm^2^ were loaded with 10 mL of 1 × 10^6^ CFU bacterial suspension. Then, the samples were further exposed to light irradiation or dark conditions for killing bacteria. Finally, the bacteria suspension was diluted for bacterial enumeration. All data were standardized as 1 × 10^6^ CFU initial load and plotted CFU.

### 2.7. Reusability Measurements 

To evaluate the reusability of the G-SF/PVA BNM, they were sonicated for 10 min and washed for 5 min to remove the attached bacteria after use. Finally, the rechargeable G-SF/PVA BNM was dried and reutilized for killing bacteria.

### 2.8. Characterization

The surface morphologies of the silk fibroin/polyvinyl alcohol blended nanofibrous membranes before and after chemical modification were observed by using a field emission scanning electron microscope (FE-SEM, Philips XL30, Hillsboro, OR, USA). Fourier transform infrared spectra (FT-IR) were recorded by a Nicolet 6700 spectrometer (Thermo Electron Co., Waltham, MA, USA). Electron spin resonance (ESR) signals were detected using a Bruker spectrometer. The mechanical strength tests of the different nanofibrous membranes were performed using an Instron Micro Tester. 

## 3. Results and Discussion

### 3.1. The Morphology and Structure

We designed the rechargeable photoactive silk fibroin/polyvinyl alcohol blended nanofibrous membranes based on four criteria: (1) ultrathin fiber diameter and good mechanical properties, (2) excellent photoactive properties and photo-store capacity, (3) good antibacterial activity, and (4) good reusability. In this work, the first goal was satisfied by using the electrospinning method. To achieve other goals, the silk fibroin/polyvinyl alcohol blended nanofibrous membranes were modified with BTDA. Finally, the rechargeable photoactive silk fibroin/polyvinyl alcohol blended nanofibrous membranes were constructed as antibacterial material for killing bacteria.

The morphological studies were performed on the silk fibroin/polyvinyl alcohol blended nanofibrous membranes (SF/PVA BNM) before and after chemical modification. The representative FE-SEM image indicated that the physical structure of the SF/PVA BNM remained stable after grafting BTDA, which is clearly demonstrated in Figure 2a,b. The blended nanofibrous membranes possessed the uniform and highly porous characteristics without droplets or beads. In addition, the diameter distributions, demonstrated from Figure 2c,d, exhibited an average diameter of approximately 285 nm and 298 nm. There is a slight increase in fiber diameter after chemical modification, which was due to the permeation of solution into fibers, leading to a slight swelling of the fibers. 

The mechanical properties of the different nanofibrous membranes were examined and compared in this work. It is shown from Figure 3 that the pure silk fibroin nanofibrous membranes had poor tensile strength (177.6 N/cm^2^) and elongation at break (5.35%) due to the weaker flexibility of the silk fibroin molecules. After the addition of polyvinyl alcohol, the mechanical properties of the nanofibrous membranes were increased obviously. The tensile strengths of the SF/PVA BNM and G-SF/PVA BNM reached 466.4 and 470.6 N/cm^2^. The elongations at break were 37.6% and 38.2%, respectively. These increases in mechanical performance were obtained for more developed *β*-sheet conformations, which were beneficial to their further applications [35,36]. 

The FTIR spectra of the SF/PVA BNM and G-SF/PVA BNM were measured and compared as shown in Figure 4. After chemical modification, there is an obvious change in the peak at 1586 cm^−1^ due to the carboxylate ions. Additionally, the main characteristic peaks of the G-SF/PVA BNM at 1722 cm^−1^ were observed, indicating the bonding of ester bonds between the BTDA and nanofibrous membranes. We believe that the successful combination of the BTDA on the blended membranes would provide the possibility for the G-SF/PVA BNM to kill bacteria. 

### 3.2. Contact Killing of Bacteria

The principle of photoactive and photo-store capacities was reported based on a previous study [37]. As shown in Figure 5a, the initial reaction happened by intersystem crossing (ISC), which was caused by the formation of a triplet excited state. After that, the triplet excited state abstracted the hydrogen atom, leading to the formation of a hydroquinone radical, which could be reacted with the oxygen molecules and form the ROS. If the hydroquinone radical is not completely reacted with oxygen, a structure rearrangement may be caused. Then, the metastable structures of the hydroquinone radical formed, which could store the activity. After reaction with oxygen, it would produce an ROS in a dark place [36]. As shown in Figure 5b,c, the concentration of OH• and H_2_O_2_ was increased in an irradiation environment and stopped in dark conditions. After irradiation, the release of the ROS was increased continuously without obvious decrease, indicating the excellent photoactivity of the G-SF/PVA BNM. In addition, the generated •OH radicals for the G-SF/PVA BNM at different time intervals were detected by a DMPO-trapping ESR technique. As displayed in Figure 5d, the ESR spectra possessed a 1:2:2:1 quartet pattern, and the intensity of the peaks was enhanced with the increasing reaction, which confirmed the formation of the •OH radicals [38,39]. To evaluate the antibacterial activity of the G-SF/PVA BNM, typical pathogenic bacteria, *Escherichia coli* (*E. coli*) and *Staphylococcus aureus* (*S. aureus*), were employed to examine the photoactive membranes. For the antibacterial experiment, the G-SF/PVA BNM with a size of 2 × 2 cm^2^ were loaded with 10 mL of 1 × 10^6^ CFU bacterial suspension. The bacterial proliferation was evaluated by agar plate counting. Figure 5e,f demonstrates the antibacterial capacity of the G-SF/PVA BNM under light irradiation. The result showed that the G-SF/PVA BNM had a good biocide function for a 6 log CFU reduction of *E. coli* and *S. aureus*, indicating its 99.9999% biocide efficacy. It is worth noting that the control group showed significant bacterial growth after 120 min. To further study the bactericidal mechanism of the G-SF/PVA BNM, the morphology of the bacteria contacting with the control group and charged group were investigated. As displayed in Figure 5f, the *E. coli* and *S. aureus* for the control group kept their intact cell membranes, while the destruction of bacterial cell walls was observed for the charged group, indicating the robust antibacterial activity of the charged group.

### 3.3. Rechargeable Property of the G-SF/PVA BNM

The photoactive properties of the G-SF/PVA BNM were further demonstrated by their promising rechargeable biocidal functions. To demonstrate this, we irradiated the G-SF/PVA BNM under light irradiation for 1 h and measured the released amount of the ROS under dark conditions. As shown in Figure 6a,b, the amount of OH• and H_2_O_2_ was rapidly increased in the first 10 min, and then it attained equilibrium. This evidence indicated that the G-SF/PVA BNM possessed good rechargeable properties, which could store photoactive activity and release it for killing bacterial under dark conditions. The G-SF/PVA BNM was charged for 1 h under light irradiation and was used for killing *E. coli* and *S. aureus* under dark conditions. As displayed in Figure 6c, the charged G-SF/PVA BNM possessed good biocidal capacity for a 6 log CFU reduction of *E. coli* and *S. aureus* in 120 min. Subsequently, the leakage of the nucleic acid of the bacteria was measured for the control group and charged group, respectively. As seen in Figure 6d, that there was nearly no organic matter detected for the control group, while there was significant leakage of nucleic acids measured for the charged group. These results indicated that the charged G-SF/PVA BNM group possessed an excellent photo-store capacity for killing bacterial, even under dark conditions [40].

### 3.4. Cyclic Performance

To examine economic cost, it is very necessary to evaluate reusability. We carried out the reversibility of the G-SF/PVA BNM for killing *E. coli*. Figure 7a shows that the excellent photoactive activity of the G-SF/PVA BNM remained after five cycles. Observation of the SEM image indicates that the structures of the blended nanofibrous membranes did not undergo obvious changes after five cycles, as displayed in Figure 7b. These results indicated that the prepared G-SF/PVA BNM would greatly improve the practical application for killing bacteria.

## 4. Conclusions

In conclusion, we have demonstrated an effective method for the construction of rechargeable photoactive silk fibroin/polyvinyl alcohol blended nanofibrous membranes (G-SF/PVA BNM) under mild conditions, which could generate reactive oxygen species (ROS) for killing bacteria under light irradiation. The G-SF/PVA BNM can store photoactive activity under light irradiation and release ROS for antibacterial applications, even under dark conditions. The resultant G-SF/PVA BNM possessed an ultrathin fiber diameter (298 nm), excellent photoactive activity and photo-store capacity, and great photoinduced antibacterial activity against *E. coli* and *S. aureus* (99.999% bacterial reduction with 120 min). The successful synthesis of antibacterial membranes can give new insights to the development of photoactive membrane materials for different applications.

## Figures and Tables

**Figure 1 polymers-14-02499-f001:**
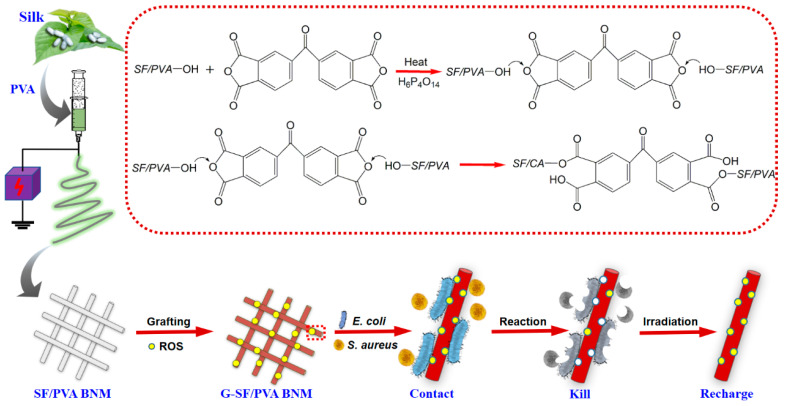
Scheme of the design, preparation, and antibacterial activity of G-SF/PVA BNM.

**Figure 2 polymers-14-02499-f002:**
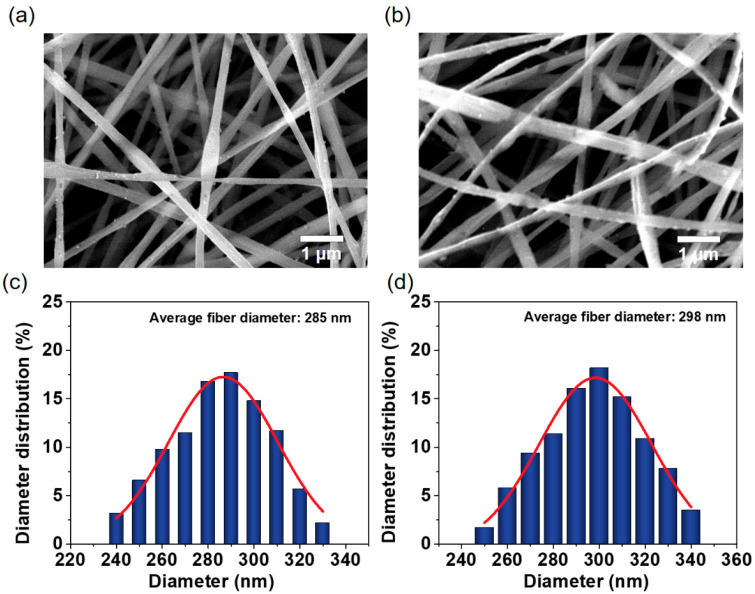
SEM images of (**a**) SF/PVA BNM and (**b**) G-SF/PVA BNM. Diameter distribution of (**c**) SF/PVA BNM and (**d**) G-SF/PVA BNM.

**Figure 3 polymers-14-02499-f003:**
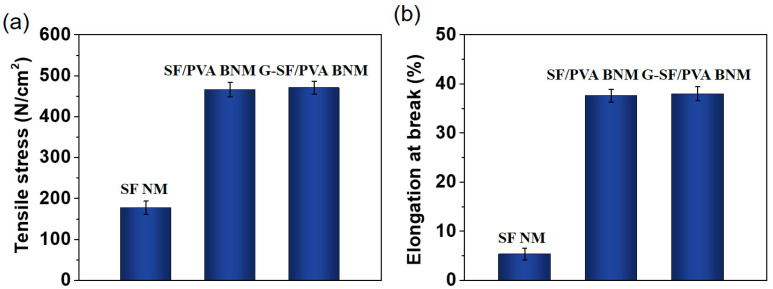
(**a**) The measurements of tensile stress. (**b**) The measurements of elongation at break.

**Figure 4 polymers-14-02499-f004:**
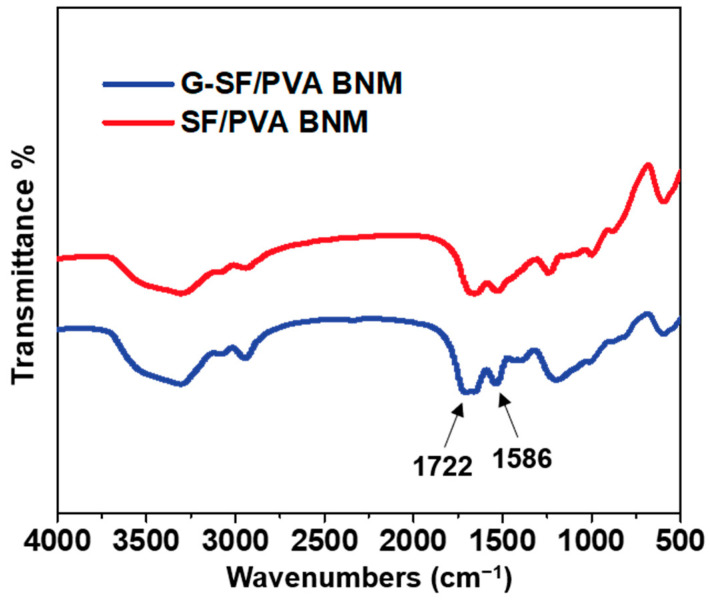
FT-IR spectra of the SF/PVA BNM and G-SF/PVA BNM.

**Figure 5 polymers-14-02499-f005:**
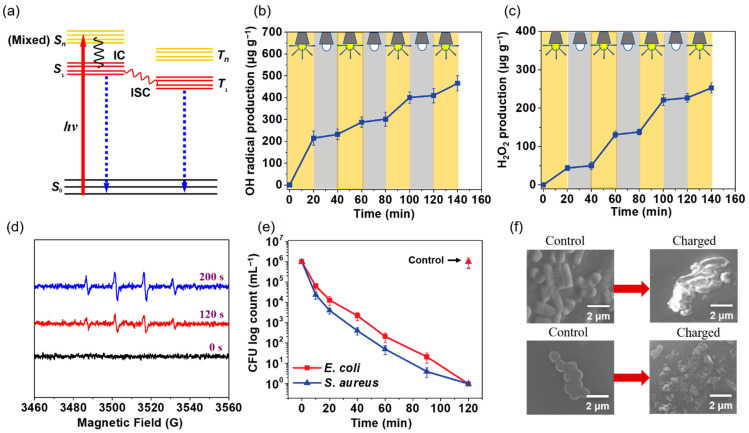
(**a**) Jablonski diagrams illustrating the photoexcitation process. (**b**) The released amount of OH• at different times. (**c**) The released amount of H_2_O_2_ at different times. (**d**) The ESR signals for the G-SF/PVA BNM at different times. (**e**) The biocidal activity against *E. coli* and *S. aureus* of the G-SF/PVA BNM in irradiated conditions. (**f**) FE-SEM images of *E. coli* and *S. aureus* cells for the G-SF/PVA BNM.

**Figure 6 polymers-14-02499-f006:**
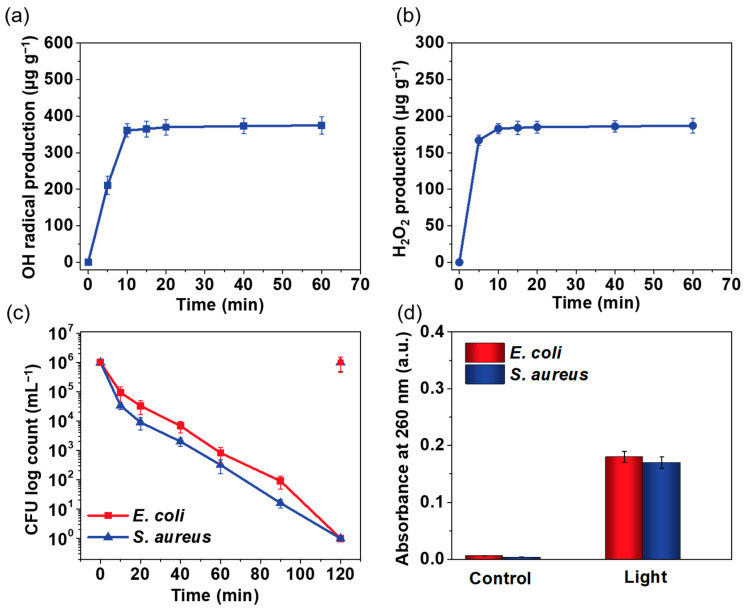
The respective concentrations of (**a**) OH• and (**b**) H_2_O_2_ produced by the G-SF/PVA BNM under dark conditions. (**c**) The biocidal activity against *E. coli* and *S. aureus* of the G-SF/PVA BNM under dark conditions. (**d**) Measurements of the leakage of nucleic acid from the *E. coli* cells and *S. aureus* cells.

**Figure 7 polymers-14-02499-f007:**
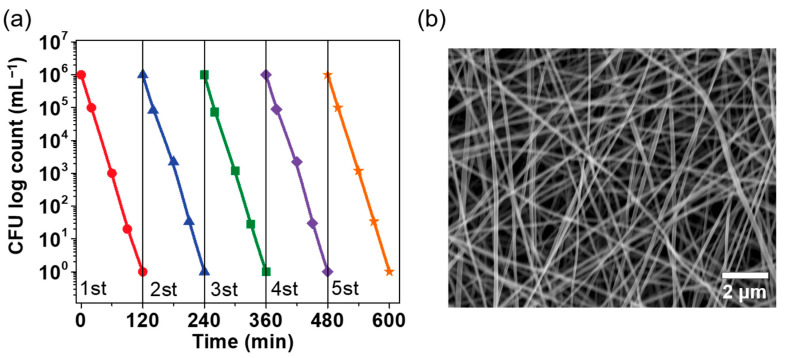
(**a**) Biocidal activity of the G-SF/PVA BNM through five cycles. (**b**) SEM images of the G-SF/PVA BNM after five cycles.

## Data Availability

The data presented in this study are available in the article.

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
