# Peer review of "Fabrication of Rechargeable Photoactive Silk Fibroin/Polyvinyl Alcohol Blend Nanofibrous Membranes for Killing Bacteria"

_polymers, 2022, doi:10.3390/polym14122499_

Round 1

Reviewer 1 Report

This manuscript SF/PVA/ blends nanofibrous membranes are fabricated and their antibacterial and photoactive properties under UV light explored. This manuscripts depicts novelty in terms of application and develop interest for the readers. However, there are some suggestion or recommendation to incorporate before publication.

1. As minor, in the whole manuscript the numbering of heading and subheadings are missing.

2. As minor, in the abstract section line 21 the strains should be italic.

3. As major, in the introduction section there need some more referenced work related to this research. The introduction seems incomplete.

4. In line 84-88 in the methodology section, please mention the standard antibacterial test method you applied. further please provide the standard bacterial strains that you have tested.

5. In line 89, Reusability measurements, please provide the standard method of washing to access the durability.

6. In figure 4, the font size of X and y axis is much higher and looks odd, please make it compatible with the manuscript font size.

Author Response

Responds to the reviewer’s comments:

Response to Reviewer 1

We thank the reviewer for the comments on our manuscript. The comments are very helpful in improving our manuscript. We have revised the manuscript according to the reviewer’s suggestions and carefully addressed all of the questions the reviewer raised. Our responses to the reviewer’s comments are listed below.

  1. Comments:

As minor, in the whole manuscript the numbering of heading and subheadings are missing.

Response:

Thank the reviewer for the comments. The numbering of heading and subheadings were added in the revised manuscript.

  1. Comments:

As minor, in the abstract section line 21 the strains should be italic.

Response:

 Thank the reviewer for the comments. It was revised to be italic in the revised manuscript.

  1. Comments:

As major, in the introduction section there need some more referenced work related to this research. The introduction seems incomplete.

Response:

We have improved the introduction section. The relevant part was revised in the manuscript.

  1. Comments:

In line 84-88 in the methodology section, please mention the standard antibacterial test method you applied. further please provide the standard bacterial strains that you have tested.

Response:

The antibacterial activity was evaluated against E. coli and S. aureus. Firstly, the G-SF/PVA BNM with a size of 2 × 2 cm2 were loaded with 10 ml of 1 × 106 CFU bacterial suspension. Then, samples were further exposed on light irradiation or dark conditions for killing bacteria. At each time point, the samples with the bacteria were harvested by vortexing with 1 ml of deionized (DI) water, and the suspension was serially diluted for bacterial enumeration. All data were standardized as 1 × 106 CFU initial load and plotted CFU. The relevant parts were added in the revised manuscript.

  1. Comments:

In line 89, Reusability measurements, please provide the standard method of washing to access the durability.

Response:

For the cyclic antimicrobial assays, after each test, the samples were sonicated for 10 min and washed for 5 min to remove the attached bacterial debris. Finally, the rechargeable G-SF/PVA BNM was dried and reutilized for killing bacteria. The relevant parts were added in the revised manuscript.

  1. Comments:

In figure 4, the font size of X and y axis is much higher and looks odd, please make it compatible with the manuscript font size.

Response:

Thank the reviewer for the comments. Figure 4 was revised in the manuscript.

We tried our best to improve the manuscript and made some changes in the manuscript. These changes will not influence the content and framework of the paper. We appreciate for Editors/Reviewers’ warm work earnestly, and hope that the correction will meet with approval.

Once again, thank you very much for your comments and suggestions.

Best regards!

Yours sincerely

Shixiong Yi

State Key Laboratory of Silkworm Genome Biology, College of sericulture, textile and biomass science, Southwest University, Chongqing, 400715, P.R. China

Reviewer 2 Report

The manuscript entitled, “Fabrication of Rechargeable Photoactive Silk Fibroin/Polyvinyl Alcohol Blend Nanofibrous Membranes for Killing Bacteria” describes the fabrication of photoactive silk fibroin/polyvinyl alcohol blend nanofibrous membranes grafted with 3,3',4,4'-benzophenone tetracarboxylic dianhydride (G-SF/PVA BNM) were fabricated by electrospinning technique. The obtained results showed that the G-SF/PVA BNM exhibited integrated properties of ultrathin fiber diameter (298 nm), good mechanical properties, robust photoactive activity and photo-store capacity, great photoinduced antibacterial activity against bacterial reduction. The following issues are needed to be addressed before consideration of this work for publication in Polymers Journal.

The author should provide the detailed experimental procedure for Electrospinning of SF-PVA nanofibers, such as, ratio measurement, needle size, flow rate, voltage power supply, and needle and collector distance.

Mechanical Properties of membrane procedure should provide in detail.

In figure 4. FT-IR spectra of SF/PVA BNM and G-SF/PVA BNM text size should be reduced in the journal format.

In Figure 3. (a) The measurements of tensile stress. (b) The measurements of elongation at break triplicate error values should be provide.

There is no reference in the results and discussion. The author should cite the suitable references in results and discussion sections.

Author Response

Responds to the reviewer’s comments:

Response to Reviewer 2

Thank you very much for your comments on our manuscript. We have revised the manuscript according to your sincere suggestions, the responses to comments are listed as follows.

  1. Comments:

The author should provide the detailed experimental procedure for Electrospinning of SF-PVA nanofibers, such as, ratio measurement, needle size, flow rate, voltage power supply, and needle and collector distance.

Response:

The detailed experimental procedure for Electrospinning of SF-PVA nanofibers was added in ‘2.2. Electrospinning of SF-PVA nanofibers’ in revised manuscript.

  1. Comments:

Mechanical properties of membrane procedure should provide in detail.

Response:

Thank the reviewer for the comment. The detail experimental method of mechanical properties of membrane was added in the revised manuscript.

  1. Comments:

In figure 4. FT-IR spectra of SF/PVA BNM and G-SF/PVA BNM text size should be reduced in the journal format.

Response:

Thank the reviewer for the comments. Figure 4 was revised in the manuscript.

  1. Comments:

In Figure 3. (a) The measurements of tensile stress. (b) The measurements of elongation at break triplicate error values should be provide.

Response:

Figure 3 was revised in the manuscript.

  1. Comments:

There is no reference in the results and discussion. The author should cite the suitable references in results and discussion sections.

Response:

We have cited some suitable references in results and discussion sections. They were added in the manuscript.

We tried our best to improve the manuscript and made some changes in the manuscript. These changes will not influence the content and framework of the paper. We appreciate for Editors/Reviewers’ warm work earnestly, and hope that the correction will meet with approval.

Once again, thank you very much for your comments and suggestions.

Best regards!

Yours sincerely

Shixiong Yi

State Key Laboratory of Silkworm Genome Biology, College of sericulture, textile and biomass science, Southwest University, Chongqing, 400715, P.R. China
